# Chemodiversity and Anti-Leukemia Effect of Metabolites from *Penicillium setosum* CMLD 18

**DOI:** 10.3390/metabo13010023

**Published:** 2022-12-23

**Authors:** Ana Calheiros de Carvalho, Cauê Santos Lima, Heron Fernandes Vieira Torquato, André Tarsis Domiciano, Sebastião da Cruz Silva, Lucas Magalhães de Abreu, Miriam Uemi, Edgar Julian Paredes-Gamero, Paulo Cezar Vieira, Thiago André Moura Veiga, Lívia Soman de Medeiros

**Affiliations:** 1Programa de Pós-Graduação em Biologia Química, Instituto de Ciências Ambientais, Químicas e Farmacêuticas, Universidade Federal de São Paulo, Diadema 09972-270, Brazil; 2Departamento de Bioquímica, Universidade Federal de São Paulo, São Paulo 04044-020, Brazil; 3Faculdade de Farmácia, Centro Universitário Braz Cubas, Mogi das Cruzes 08773-380, Brazil; 4Instituto de Ciências Exatas, Universidade Federal do Sul e Sudeste do Pará, Marabá 68505-080, Brazil; 5Departamento de Fitopatologia, Universidade Federal de Viçosa, Viçosa 36570-900, Brazil; 6Departamento de Química, Instituto de Ciências Ambientais, Químicas e Farmacêuticas, Universidade Federal de São Paulo, Diadema 09972-270, Brazil; 7Faculdade de Ciências Farmacêuticas, Alimentos e Nutrição, Universidade Federal de Mato Grosso do Sul, Campo Grande 79070-900, Brazil; 8NPPNS, Department of BioMolecular Sciences, Faculty of Pharmaceutical Sciences of Ribeirao Preto, University of Sao Paulo, Ribeirao Preto 14040-903, Brazil

**Keywords:** *Penicillium setosum*, *Swinglea glutinosa*, dereplication, leukemia

## Abstract

*Penicillium setosum* represents a *Penicillium* species recently described, with little up-to-date information about its metabolic and biological potential. Due to this scenario, we performed chemical and biological studies of *P. setosum* CMLD18, a strain isolated from *Swinglea glutinosa* (Rutaceae). HRMS-MS guided dereplication strategies and anti-leukemia assays conducted the isolation and characterization of six compounds after several chromatographic procedures: 2-chloroemodic acid (**2**), 2-chloro-1,3,8-trihydroxy-6- (hydroxymethyl)-anthraquinone (**7**), 7-chloroemodin (**8**), bisdethiobis(methylthio)acetylaranotine (**9**), fellutanine C (**10**), and 4-methyl-5,6-diihydro-2H-pyran-2-one (**15**). From the assayed metabolites, (**10**) induced cellular death against Kasumi-1, a human leukemia cell line, as well as good selectivity for it, displaying promising cytotoxic activity. Here, the correct NMR signal assignments for (**9**) are also described. Therefore, this work highlights more detailed knowledge about the *P. setosum* chemical profile as well as its biological potential, offering prospects for obtaining natural products with anti-leukemia capabilities.

## 1. Introduction

The *Penicillium* genus has a worldwide distribution, including 483 species reported so far. Some of them are known to have a tremendous economic impact on human life due to the production of bioactive metabolites [1]. The genus is notable as one of the main producers of natural products approved by the FDA, with applications in several areas, such as medicine and biotechnology. *Penicillium* species provide an important source of promising bioactive compounds [2,3], including metabolites with anticancer properties [2,3], such as wortmannin, 3-*O*-methylfunicone, dicatenarin, and 3-*O-*methylfunicone [4]. Additionally, there are some examples of compounds with effects against human leukemia cell lines, such as emodin, brefeldin A, griseofulvin, and mycophenolic acid, with polyketides and alkaloids being the most representative classes [2].

The species *P. setosum* was first reported by George et al. (2019) as an endophytic fungus from *Withania somnifera* (L.) Dunal [5]. The production of the mycotoxins patulin, chloromycetin, and andrastin D was described for the species, as well as some flavonoids such as quercetin, dihydroquercetin, dihydromyricetin, kaempferol, and luteolin [6]. Some other polyketides, hydrocarbons, and fatty acids have also been reported [6]. However, all these metabolites have been pointed out only on the basis of high-resolution mass spectrometry data analysis in crude extracts and manual fragmentation study for some of the described compounds [6]. Any metabolite isolation and further chemical characterization or insights about *P.setosum* chemodiversity bioactivity properties have been provided since then.

Quite often, mixtures of natural products are very complex, representing a challenge for compound identification, leading to costs and time-consuming workflows. In order to overcome these drawbacks, dereplication strategies based on mass spectrometry data analysis can provide faster results regarding metabolic fingerprinting. These data can be organized in molecular clusters based on MS/MS fragmentation studies [7]. Among the strategies, the GNPS platform should be cited as a notable example that allows MS users to accelerate the recognition of known substances through the creation of molecular networks by eliminating onerous purification steps [8]. 

Given the lack of further data on this *Penicillium* species’ secondary metabolism, we decided to explore the chemical profile and the biological potential of *P. setosum* CMLD 18, isolated as an endophyte from *Swinglea glutinosa* (Rutaceae). To accelerate the access to its chemical fingerprinting, we applied MS/MS dereplication tools, including molecular networking data analysis, while the activity against leukemia cell lines from the fractions and isolated compounds were also evaluated. The combination of these approaches led to the annotation of fifteen metabolites and the isolation of six of them. Therefore, this investigation allowed us to describe substances unrelated to the *Penicillium* genus in addition to evaluating their cytotoxic potential. 

## 2. Experimental Design

### Equipment

The NMR spectra were recorded on a Bruker spectrometer (Bruker Daltonics^®^, Billerica, MA, USA) Ultrashield 300—Advance III operating at 300 MHz (^1^H) and 75 MHz (^13^C). UHPLC-HR-ESI-MS/MS data were recorded on a Shimadzu Nexera X2 liquid chromatography system (Shimadzu) equipped with an SPD-M20A Proeminence Diode Array detector (Shimadzu), using a reverse-phase Kinetex, C18 (2.6 μ—100.0 mm × 2.1 mm—Phenomenex^®^, Torrance, CA, USA). The LC system was coupled to a quadrupole time-of-flight mass Bruker Esquire 3000 Plus device (Bruker) on a hybrid equipment type QTOF (MicroTof-QII, Bruker), equipped with ESI operating in positive and negative ion mode.

For MS/MS dereplication via molecular networking analysis, MS/MS data were acquired using AutoMS mode and converted to .mzXML format using MS-Convert software, which is part of ProteoWizard (Palo Alto, CA, USA). The networks were generated using the online platform (https://gnps.ucsd.edu/ProteoSAFe/static/gnps-splash.jsp), accessed on 1 December 2022 [9]. All MS/MS peaks within ±17 Da deviations from the precursor ions were filtered out. MS/MS spectra were selected from only the six best peaks, considering a range of ±50 Da across the spectrum. The data were grouped with a tolerance of 0.02 Da for precursor ions and 0.02 Da for fragment ions in the construction of “consensus” spectra (identical spectra for each precursor, which are combined to create the node to be visualized). Consensus spectra with less than two spectra were not considered. Connections between nodes were filtered to values greater than 0.6 of the cosine parameter, with compatibility for more than six peaks. For the dereplication of compounds, the generated network spectra were consulted at the GNPS libraries, using the same selection criteria for the analyzed samples. GNPS data were analyzed and viewed using Cytoscape 3.7.0 software (U.S. National Institute of General Medical Sciences, Bethesda, MD, USA).

## 3. Procedure

### 3.1. Biological Assays

#### 3.1.1. Stock Solutions

Stock solutions of all compounds were prepared in DMSO, stored at −20 °C, and diluted in a culture medium before use. The final concentration of DMSO in the culture medium was not higher than 0.25% at any time. 

#### 3.1.2. Cell Cultures

Human leukemic cell lines (Kasumi-1, KG-1, MOLT-4, and Jurkat) were obtained from the American Type Culture Collection (ATCC). KG-1 cells were maintained in Iscove’s modified Dulbecco’s medium (IMDM) supplemented with 20% fetal bovine serum (FBS) (Cultilab, Campinas, Brazil), according to criteria adopted by the cell bank (https://www.atcc.org/products/ccl-246#detailed-product-information) accessed on 10 November 2021, whereas the other lineages were maintained in Roswell Park Memorial Institute medium (RPMI 1640) (Sigma Aldrich, St. Louis, MI, USA) supplemented with 10% FBS. All cells were cultured in a medium containing 100 U/mL penicillin (Sigma Aldrich, St. Louis, MI, USA) and 100 μg/mL streptomycin (Sigma Aldrich, St. Louis, MI, USA) and in a humidified incubator containing 5% CO_2_ at 37 °C. Passage numbers for all leukemic cell lines were between 3 and 6.

#### 3.1.3. Peripheral Blood Mononuclear Isolation

Peripheral blood mononuclear cells (PBMCs) were obtained from three healthy donors. Human monocytes were collected from healthy donors after informed patient consent was obtained. Separation of mononuclear cells was performed by gradient centrifugation methods using Ficoll Histopaque-1077 (1.077 g/cm^3^) (Sigma Aldrich, St. Louis, MI, USA) following the manufacturer’s instructions. The use of human samples was approved by the local Ethical Committee of the Universidade Federal de Mato Grosso do Sul (CAAE35853720.2.0000.0021). Cells were maintained in IMDM supplemented with 20% FBS, 100 U/mL penicillin, and 100 μg/mL streptomycin in a humidified atmosphere at 37 °C in 5% CO_2_.

#### 3.1.4. Resazurin Assay

A resazurin assay (Sigma Aldrich, St. Louis, MI, USA) was used to screen and assess the activity of extracts, fractions, and isolated compounds against leukemic cell lines. Briefly, 10^5^ cells/mL were seeded in 96-well plates containing supplemented medium. Cells were treated in the presence or absence of compounds for 48 h using a high concentration (100 µg/mL or 100 µM to isolated compounds) or a concentration–response curve (100, 50, 25, 12.5, 6.25, 3.12, 1.56, and 0.78 µM) after 24 h of treatment with isolated compounds was constructed. After treatment, the cell culture medium was removed, and 100 µL of medium with 10% Resazurin solution was added; after 4 h of incubation, the fluorescence was measured (excitation 530 nm, emission 590 nm) in a FlexStation 3 microplate reader (Molecular Devices, San Jose, CA, USA). Each experiment was performed in triplicate.

#### 3.1.5. Annexin V/7-AAD Flow Cytometry Assay

An Annexin V-FITC/7-AAD double staining assay was performed to evaluate the proapoptotic effect of **10**. Kasumi-1 and Jurkat cells (10^5^/mL) were treated for 24 h with EC_50_, 50, or 50 µM **10**, respectively, or Staurosporine (1 µM) (Sigma Aldrich, St. Louis, MI, USA). Then, the cells were resuspended in annexin binding buffer (0.14 M NaCl, 2.5 mM CaCl2, 0.01 M HEPES, pH 7.4) and incubated at room temperature with 1 μL and 2 μL of annexin V-FITC and 7-AAD (Becton Dickinson, Covington, GA, USA), respectively, for 30 min. Data were analyzed using FlowJo V10 (Becton Dickinson—BD, USA) software; 20,000 events were collected per sample.

#### 3.1.6. Intracellular Protein Labeling

Kasumi-1 and Jurkat cells (10^5^/mL) were treated for 24 h with EC_50_, 50, or 50 µM **10**, respectively. Then, cells were fixed with BD Cytofix (BD Biosciences, Mississauga, ON, Canadá) for 15 min, washed with BD Perm/Wash buffer, and permeabilized with Perm Buffer III (BD, USA) for 30 min at room temperature. In order to label cleaved caspase-3 protein (Cell Signaling, Danvers, MA, USA), cells were incubated for 1 h with primary antibody (1:800). Anti-rabbit IgG secondary antibodies conjugated with Alexa Fluor 488 (Thermo Fisher Scientific, Waltham, MA, USA) (1:1000) were incubated for at least 40 min, and the fluorescence was measured using an Accuri C6 flow cytometer and FlowJo software to analyze the results; 20,000 events were collected per sample. Protein analyses were performed using GM fluorescence intensity.

#### 3.1.7. Cell Cycle Analysis

The distribution of cell cycle phases was determined by propidium iodide (PI) staining and flow cytometry analysis. Kasumi-1 and Jurkat cells (10^5^/mL) were treated with **10** (10 μM or 1 μM, respectively) for 72 h. Then, the cells were fixed and permeabilized as previously described and treated with 4 μg/mL RNase (Sigma Aldrich, St. Louis, MI, USA) for 45 min at 37 °C. For DNA labeling, cells were incubated with 5 μg/mL of PI (Sigma Aldrich, St. Louis, MI, USA). Percentages of cells within cell cycle compartments (G1/G0, S, and G2/M) were performed using an Accuri C6 flow cytometer (BD Biosciences, Mississauga, Canadá). A total of 40,000 events were acquired.

#### 3.1.8. Cell Differentiation by Immunophenotyping

Cells (10^5^/mL) were treated with 10 μM Compound **10** for 72 h. Then, the cells were collected and stained with CD11b-Cy7/PE (2 μL) (BD, USA). The measurements of fluorescence intensity were performed using an Accuri C6 flow cytometer (BD Biosciences, Mississauga, Canadá); 40,000 events were collected per sample. Results were expressed as GM fluorescence intensity.

#### 3.1.9. Statistical Analyses

All data represent at least three independent experiments and are expressed as the mean ± standard error of the mean (SEM). Statistical analyses were performed using Student’s *t*-test for comparisons between two groups and analysis of variance (ANOVA) and Dunnett’s post hoc test for multiple comparisons among groups. A probability value of *p* < 0.05 vs. the control was considered significant. GraphPad Prism software V6 was used for data analyses.

### 3.2. Isolation of Penicillium setosum from Swinglea glutinosa

*Swinglea glutinosa* was harvested on the same day that the fungal isolation procedure was performed. For future reference, a voucher (HUFSP 1063) of the specimen was deposited in the Federal University of São Paulo herbarium. 

In a biological safety cabinet, the plant material was sequentially dipped in 70% ethanol, distilled water, 11% aqueous sodium hypochlorite solution, distilled water, 70% ethanol, and distilled water (2 min of each immersion) to eliminate epiphytic microorganisms and contaminating agents [10]. 

The leaves, stem, and roots of *S. glutinosa* were fragmented into small pieces (approximately 1 cm). These fragments were deposited in Petri dishes containing PDA (Potato-Dextrose-Agar) amended with the antibiotic chloramphenicol (100 mg/L). Subsequently, plates were stored in a BOD (Biochemical Oxygen Demand) incubator at 25 °C and observed for 7 days. Successive inoculation procedures were performed to purify the fungal colonies (Appendix A).

### 3.3. Fungi Cultivation and Initial Characterization 

*P. setosum* was inoculated in malt extract (2%) agar and oatmeal (3%) agar. The plates were incubated on a bench for up to 15 days, and the colonies were inspected for the detection of reproductive structures. The beta tubulin (TUB2) gene was chosen for phylogeny.

#### 3.3.1. DNA Extraction, PCR, and Sequencing

The beta tubulin (TUB2) gene was chosen for phylogenetic analyses and species identification. The fungus was cultured on plates with malt-agar extract medium for seven days; the mycelium was then scraped off, transferred to ceramic crucibles, and macerated with liquid N_2_. The macerates were transferred to Eppendorf tubes, and the genomic DNA was extracted with the Wizard^®^ Genomic DNA Purification Kit (Promega). DNA samples were subjected to PCR reactions using the primers and reaction conditions described by Stielow et al. (2015) [11]. PCR products were purified with ExoSAP-IT™ PCR Product Cleanup Reagent and sent for two-way sequencing by Macrogen Korea.

#### 3.3.2. Phylogenetic Analysis and Identification

The generated sequence (GenBank accession number OP019359) was edited in the SeqAssem program and compared with the NCBI database using the BLAST tool. Based on the results of the comparison, sequences of reference isolates from species phylogenetically close to the fungus were downloaded and used to perform multiple DNA alignments in the MEGA 7 program. 

Phylogenetic analysis was performed on the Cipres portal (www.phylo.org, accessed on 1 November 2022) using two inference methods, maximum likelihood (ML) and Bayesian inference. ML trees were estimated using the RAxML v8 program, with the option “rapid bootstrap analysis/search for the best-scoring ML tree.” Tree branch supports were estimated using 1000 pseudo-replicas of the bootstrap analysis. Bayesian inference trees were estimated in the MrBayes 3.27 program. The evolutionary HKY+G was model inferred for the beta tubulin alignment using JModeltest2. For each analysis, two runs with four strands each were repeated for up to 5,000,000 generations and sampled every 1000 generations. The analysis was terminated when the standard deviation between the bipartitions of the two runs reached the value of 0.01. Consensus trees were built after discarding 25% of the initial trees. The generated trees were visualized on the program FigTree and edited on the program InkScape (for details, see Appendix A). 

#### 3.3.3. *Penicillium setosum* Cultivation in Hominy (M1) and Rice (M2)

The fungal culture was cultivated in hominy. For extract obtention, 26 mL of distilled water was added to 30 g of hominy (Yoki^®^) in two 250 mL Erlenmeyer flasks. The flasks were autoclaved for 40 min at 120 °C and 1 atm of pressure. Five fragments (5 mm in diameter) from the fungus cultivated in PDA medium were aseptically transferred to each of the flasks containing the sterilized hominy. The flasks were incubated for 35 days at room temperature in static mode. At the end of the incubation period, 75 mL of MeOH was added to the flasks. After 12 h of extraction, all content was vacuum filtered. The extraction procedure was repeated after 12 h. After solvent elimination, the methanol extract of *P. setosum* from cultivation in hominy was obtained. 

For fungal cultivation in rice, 80 mL of distilled water was added to 90 g of rice (Uncle Ben’s). The same procedure performed for cultivation in hominy was then repeated. For cultivation in large-scale aiming compound isolation, the same procedure was repeated using 40 Erlenmeyer flasks (500 mL), each one with 90 g of rice and 80 mL of distilled water.

#### 3.3.4. *Penicillium setosum* Cultivation in CYB (M3) and YES (M4)

The fungal culture was cultivated in Czapek yeast extract broth (CYB). For extract obtention, 90 mL of CYB culture media (NaNO_3_: 3 g/L; yeast extract: 5 g/L; glucose: 30 g/L; K_2_HPO_4_.3H_2_O: 1.3 g/L; concentrated Czapeck: 10 mL/L) was added to two 500 mL Erlenmeyer flasks. The flasks were autoclaved for 40 min at 120 °C and 1 atm of pressure. Five fragments (5 mm in diameter) from the fungus cultivated in the PDA medium were then added. After 15 days of incubation at room temperature without agitation, 300 mL of MeOH was added to the Erlenmeyer flasks containing the culture medium. After 12 h of extraction, all contents were filtered under reduced pressure, separating the filtrate from the mycelium; 200 mL of MeOH was then added to the mycelium. After 12 h, the contents were filtered under reduced pressure, and liquid–liquid extraction was carried out with the filtrate using AcOEt (3 times). After solvent elimination, the methanolic extract from the mycelium and the ethyl acetate extract from the filtrate was generated.

For cultivation in yeast extract sucrose (YES) (Sucrose: 150 g/L; MgSO_4_: 0.5 g/L; ZnSO_4_: 0.01 g/L; CuSO4: 0.005 g/L; Yeast extract: 20 g/L), 90 mL of YES media was added to two 500 mL Erlenmeyer flask. It was autoclaved for 40 min at 120 °C and 1 atm of pressure. In order to prepare the YES + HBr medium (M5), HBr was added to the Erlenmeyer flask until the solution reached pH 5.5. The same procedure performed for cultivation in CYB media was then repeated.

#### 3.3.5. *Penicillium setosum* Cultivation in Yes + Agar (M6)

The YES + Agar culture media was composed of: sucrose: 150 g/L; MgSO_4_: 0.5; ZnSO_4_: 0.01 g/L; CuSO_4_: 0.005 g/L; yeast extract: 20 g/L, and agar: 15 g/L. Microorganisms were grown in a Petri dish. After inoculation, plates were stored in the dark at 25 °C. All cultures were performed at 7 and 14 days. In order to obtain the micro extracts, six 5 mm diameter plugs (five from each colony) were sectioned with the aid of a cylindrical cutter. The plugs were transferred to a 2 mL vial. In each vial, 0.6 mL of the solvent mixture MeOH:AcOEt:CH_2_Cl_2_ (1:3:2) with 0.1% formic acid was added. Vials were placed in an ultrasound bath for 60 min. At the end of this period, the supernatant was transferred to a 2 mL Eppendorf tube, followed by evaporation of the solvent. The contents were then resuspended in HPLC grade MeOH, centrifuged (15,000 rpm, 10 min), and transferred to the analysis vial by UHPLC-HRMS.

### 3.4. Compounds Isolation 

The *Penicillium setosum*’s methanolic extract (104.3 g), obtained from the cultivation in rice, was subjected to liquid–liquid partition, generating hexane (10.77 g), dichloromethane (4.92 g), and ethyl acetate (2.64 g) fractions, the last of which was chromatographed under a silica gel (70–230 mesh) column (diameter: 4.5 cm; height: 32 cm). We employed hexane, dichloromethane, ethyl acetate, and methanol as eluents in a gradient mode (100% hexane, hexane/CH_2_Cl_2_ 9:1, hexane/CH_2_Cl_2_ 8:2, hexane/CH_2_Cl_2_ 7:3, hexane/CH_2_Cl_2_ 6:4, hexane/CH_2_Cl_2_ 1:1, hexane/CH_2_Cl_2_ 4:6, hexane/CH_2_Cl_2_ 3:7, hexane/CH_2_Cl_2_ 2:8, hexane/CH_2_Cl_2_ 1:9, CH_2_Cl_2_ 100%—the same gradient was used for the other pairs of solvents, in increasing order of polarity—CH_2_Cl_2_/Ethyl acetate and Ethyl acetate/methanol). We obtained 19 subfractions named C1–C19. All of these were analyzed by analytical HPLC. It was then found that the C4 fraction was constituted by the compound (**15**, 7.6 mg). Fraction C6 was submitted to preparative HPLC (C18 Luna—Phenomenex^®^—250 × 10.0 mm, 5 μ), allowing the isolation of (**9**, 3.0 mg). The employed mobile phase was formed by acetonitrile (ACN) and H_2_O (both with the addition of 0.1% formic acid). The method used for all these substances was: 0.01–2.5 min—15% ACN; 2.5–20 min—15–95% ACN; 20–27 min—95% ACN; 27–32 min—95–15% ACN; 32–35 min—15% ACN. 

The dichloromethane fraction was also chromatographed. This fraction was put into a flash silica (230–400 mesh) column (diameter: 6.0 cm; height: 27 cm) using hexane, dichloromethane, ethyl acetate, and methanol as eluents in gradient mode, yielding 13 fractions (F1–F13). Fraction F4 was submitted to preparative HPLC (C18 Luna—Phenomenex^®^—250 × 10.0 mm, 5 μ) using gradient method ACN and H_2_O (both with the addition of 0.1% of formic acid). The employed method for isolation of substances (**10**, 22.4 mg) and (**8**, 9.4 mg) was: 0.01–2.5 min—60% ACN; 2.5–12 min—60–95% ACN; 12–20 min—95% ACN; 20–23 min—95–60% ACN; 23–27min—60% ACN.

Additionally, fraction F9 was also submitted to a silica (70–230 mesh) column (diameter: 2.8 cm and height: 30 cm), using mixtures of hexane, ethyl acetate, and methanol as eluents to generate the subfractions G1-G6, with G5 being constituted by (**7**, 4.6 mg). G6 fraction (130 mg) was placed into another silica column (diameter: 2.8 cm and height: 28 cm) using hexane, ethyl acetate, and methanol as eluents, generating four fractions named H1-H4. Fraction H3 (50 mg) was purified with preparative HPLC (C18 Luna—Phenomenex^®^—250 × 10,0 mm, 5 μ) under an isocratic method composed of ACN and H_2_O (1:1), both with the addition of 0.1% of formic acid. This procedure led us to isolate compounds (**7**, 4.6 mg) and (**2**, 3.0 mg).

2-Chloroemodic acid (**2**): ^1^H NMR (CDCl_3_), δ (J/Hz): 8.26 (H-5); 7.73 (H-7); 7.17 (H-4). ^13^C NMR (CDCl_3_): 133.8 (H-10a); 125.2 (C-7); 120.9 (C-5); 114.8 (C-2). HRMS: *m/z* [M-H]^−^ 332.9817 (calcd for C_15_H_6_ClO_7_, ∆ 4.5 ppm).

2-Chloro-1,3,8-trihydroxy-6-(hydroxymethyl)-anthraquinone (**7**): ^1^H NMR (Acetone-d_6_), δ (J/Hz): 12.46 (OH); 11.77 (OH); 7.19 (H-5, *s*), 6.78 (H-7, *s*); 6.67 (H-4, *s*); 6.07 (OH, *s*); 5.10 (OH, *s*); 4.15 (CH_2_, *d*). ^13^C NMR (DSMO-d_6_): 181,98 (C-10); 161.21 (C-8); 160.03 (C-1); 152.10 (C-6); 132.90 (C-10a); 132.10 (C-4a); 120.76 (C-5); 116.85 (C-8a); 114.60 (C-2); 112.59 (C-4); 62.07 (CH_2_). HRMS: *m/z* [M-H]^-^ 319.0024 (calcd for C_15_H_8_ClO_6_, ∆ 4.5 ppm)

7-Chloroemodin (**8**): ^1^H NMR (CDCl_3_), δ (J/Hz): 12.00 (OH, *s*); 7.56 (H-4, *s*); 7.42 (H-5, *s*); 7.15 (H-2, *s*); 2.30 (*s*). ^13^C NMR (CDCl_3_): 182.0 (C-10); 149.8 (C-3); 121.6 (C-4); 109.7 (C-5); 22.0 (Me). HRMS: *m/z* [M-H]^-^ 303.0080 (calcd for C_15_H_8_ClO_5_, ∆ 6.5 ppm)

Bisdethiobis(methylthio)acetylaranotine (**9**): ^1^H NMR (CDCl_3_), δ (J/Hz): 6.57 (H-5, *d*, 8.2); 6.29 (H-7, *dd*, 8.3 and 2.2); 5.80 (H-9, *dl*, 8.1); 5.17 (H-10, *dl*, 7.6); 4.69 (H-8, *dd*, 8.2 and 1.3), 3.03 (H-3, *m*); 2.26 (H-12, *s*), 2.07 (H-14, *s*). ^13^C NMR (CDCl_3_): 170.2 (C-13), 164.5 (C-1); 139.8 (C-7); 137.8 (C-5); 109.6 (C-4), 105.9 (C-8); 71.9 (C-9); 70.6 (C-2); 60.4 (C-10); 40.6 (C-3); 21.1 (C-14); 14.8 (C-12). HRMS: *m/z* 557.1028 [M+Na]^+^ (calcd for C_24_H_26_N_2_O_8_S_2_Na^+^, ∆ 0.06 ppm).

Fellutanine C (**10**): ^1^H NMR (CDCl_3_), δ (J/Hz): 8.19 (1-NH); 7.53 (H-7, *d*, 7.6); 7.34 (H-4, *d*, 7.7); 7.16 (H-5, *m*); 7.16 (H-6, *m*); 6.16 (H-15, *dd*, 17.5 and 10.4); 5.78 (13-NH); 5.23 (H-16b, *d*, 3.4); 5,18 (H-16a, *d*, 2.4); 4,40 (H-11, *dl*, 11.3); 3.75 (H-10b, *dd*, 14.1 and 3.3); 3.27 (H-10a, *dd*, 11.8 and 14.5); 1.59 (H-14a, *s*); 1.58 (H-14b, *s*). ^13^C NMR (CDCl_3_): 168.0 (C-12); 147.5 (C-15); 142.5 (C-2); 136.1 (C-8); 130.2 (C-9); 121.8 (C-6); 119.8 (C-7); 111.9 (C-16), 111.8 (C-4); 111.7 (C-5); 106.1 (C-3); 57.2 (C-11); 39.9 (C-14); 32.5 (C-10); 28.7 (C-14b); 28.6 (C-14a). HRMS: *m/z* 507.2750 [M-H]^−^ (calcd for, ∆ 2.0 ppm). [α]_20D_ = −79 (c 0.1, MeOH).

4-Methyl-5,6-diihydro-2H-pyran-2-one (**15**): ^1^H NMR (CDCl_3_), δ (J/Hz): 5.75 (H-2, *q*); 4.32 (H-5, *t*, 6.3); 2.34 (H-4, *t*, 6.3); 1.95 (Me, *s*). ^13^C NMR (CDCl_3_): 164.7 (C-1); 158.1 (C-3); 116.6 (C-2); 65.9 (C-5); 29.2 (C-4); 23.0 (Me). HRMS: *m/z* 113,0601 [M+H]^+^ (calcd for C_6_H_9_O_2_, ∆ 1.4 ppm).

## 4. Results

### 4.1. Exploring the Chemodiversity of P. setosum 

In order to verify how abiotic conditions [12] may induce changes in the chemical profile of *P. setosum*, it was cultivated in six different culture media: hominy (M1), rice (M2), CYB (M3), YES liq (M4), YES + HBr liq (M5), and YES + Agar (M6). Methanolic micro-extracts were obtained from each medium, followed by analysis through UHPLC-HRMS. Extracts from M1 and M2 presented activity against the acute myeloid cell line (discussed in the section below). The MS data referring to the generated extracts were submitted to the GNPS platform [8] with the aim of investigating their chemical profiles. This approach allowed us to annotate fifteen compounds produced in those culture media (Table 1). Of the molecular families observed, the cluster comprised of xanthones and anthraquinones is presented in Figure 1, as well as their chlorinated derivatives. 

Compounds highlighted in yellow represent the chlorinated substances annotated through the GNPS; in red are those not annotated by the platform but identified as chlorinated substances because of their isotopic pattern (Appendix A). Green represents the nodes related to the non-chlorinated compounds suggested by the platform, and finally, in blue, there are the annotated compounds that do not have an expected fragmentation pattern for chlorinated substances (Figure 1). 

In the bioactive methanolic fraction obtained from M2, three of those chlorinated compounds were detected, two of which were annotated by the GNPS. Regarding unannotated compounds, a search for accurate masses was performed in the Dictionary of Natural Products and Antibase (2012) databases, as well as in the current literature referring to the *Penicillium* genus. For *m/z* 303.008, the substances 5-chloroemodine and 7-chloroemodine were found (error of ± 5ppm). The other chlorinated compounds present in this extract were annotated as 2-chloro-1,3,8-trihydroxy-6-hydroxymethyl-9,10-anthracenedione (**7**) (*m/z* 319.011) and 2-chloro-1,3,8-trihydroxy-2-carboxylicacid-9,10-dioxoanthracene (**2**) (*m/z* 332.993), as shown in Figure 1. It is important to note that it was also possible to observe compounds with a typical pattern for chlorination, which was not annotated in the GNPS platform and not found in the databases (Figure 1, red nodes). 

Diketopiperazine’s molecular family was also observed in the molecular network. Fellutanine C (**10**) (*m/z* 507.2750) was identified during the dereplication process using the Dictionary of Natural Products database (https://dnp.chemnetbase.com/), accessed on 5 August 2022. However, bisdethiobis(methylthio)acetylaranotine (**9**) (*m/z* 557.1028) was not found in any consulted database. After the employment of the purification and characterization procedures described below, its structure was defined. Additionally, these metabolites were found in a molecular family where no compounds were annotated (Figure 2). The molecular masses were also searched for in the cited databases above and were not found, which suggests that *P. setosum* may produce unrelated substances belonging to these classes. Based on the perspectives gained from obtaining the mentioned compounds and our search for anti-leukemia metabolites, we performed large-scale cultivation of *P. setosum* in rice once its extract (M2) was found to be considered cytotoxic against acute myeloid cell lines (Appendix A). Furthermore, canrenone (**11**), tricin (**12**), 5,6,2′-trimethoxyflavone (**13**), and hydroquinidine (**14**) were also annotated (Table 1) with the platform. The flavonoids previously reported [6] were not observed in our culture media.

### 4.2. The Isolation and Characterization of Metabolites from P. setosum

The methanolic extract was subjected to chromatographic procedures, such as silica column chromatography and high-pressure liquid chromatography, which led to the isolation of three chlorinated compounds: 7-chloroemodin (**8**) [13], 2-chloroemodic acid (**2**) [14], and 2-chloro-1,3,8-trihydroxy-6-(hydroxymethyl)-anthraquinone (**7**) [14,15]. Their chemical structures were characterized by 1D and 2D NMR data and by comparison with previously reported data (Appendix A, respectively, Appendix A). The purification of 7-chloroemodin (**8**), initially annotated as 5-chloroemodin, was very important once we could unequivocally identify its structure. Additionally, from the same fraction, we also purified another three substances not annotated by the GNPS platform: 4-methyl-5,6-diihydro-2H-pyran-2-one (**15**) [16], bisdethiobis(methylthio)acetylaranotine (**9**) [17], and fellutanine C (**10**) [18] (Appendix A, respectively, Appendix A), being (**9**) unrelated to the *Penicillium* genus.

While analyzing the spectroscopic data, we noticed some divergencies from our analysis and the data referring to compound (**9**). Nagarajan et al. (1968) reported (**9**), for the first time, isolated from *Arachniotus aureus* (Eidam) [19]. In 2009, Wang et al. isolated this substance from a marine-derived fungal species, *Alternaria raphani*. Gao et al. (2017) [17] also reported the isolation of this substance from *Aspergillus sydowii*. In the last two reports, we believe that the structural identification of (**9**) was performed probably by comparison with data reported by Nagarajan, R. et al. (1968) [19]. However, during our structural characterization process, through COSY and HMBC experiments (Appendix A), we observed some differences in the NMR shifts attributed to the first report.

According to the literature [18], at the ^1^H NMR data of (**9**), a sign at δ 6.57 was attributed to position H-7, while δ 6.29 was assigned to H-5. However, through our analysis of the COSY experiment, the sign at δ 6.29 corresponded to the hydrogen that couples with H-8 (δ 4.69) and H-9 (δ 5.80). Therefore, we believe this sign should be correctly assigned to H-7. Corroborating this proposition, in the HMBC experiment, we observed that H-7 (δ 6.29) presented a long-range correlation with C-8 and C-9 (Figure 3A). 

Another divergence observed was the assignment of the chemical shifts attributed to H-12 and H-14, whose reported values were δ 2.07 and δ 2.26, respectively. In our analysis, we assigned δ 2.26 to H-12 (Table 2). Based on its chemical shift value, we concluded that this group is connected to sulfur. HRMS analysis was extremely important for clarifying this once the pseudomolecular ion with *m/z* 557.1028 [M+Na]^+^ was observed to correspond to the suggested molecular formula C_24_H_26_N_2_O_8_S_2_Na^+^. As shown in Appendix A, its isotopic pattern was consistent with the simulation performed for this molecular formula (through the IsotopePattern—data analysis function). Additionally, in the HMBC experiment, we observed a correlation of hydrogen at δ 2.07 with the carbonyl at C-13. Therefore, this suggests that the signal at δ 2.07 represents the methylic group connected at C-14. In support of our hypothesis, the HMBC experiment confirmed the connectivity of the methylthio group, as a correlation between the methyl hydrogens H-12 (δ 2.26) and C-2 (Figure 3B) was observed.

Likewise, the signs at δ 71.9 and 70.6 were attributed to C-2 and C-9, respectively. These assignments are also divergent from what our experiments revealed. By observing our HSQC and HMBC data (Appendix A), it can be observed that the carbon at δ 71.9 presented a correlation between 3J and H-7. Furthermore, in the HSQC map, we observed that the signal at δ 70.6 did not present a direct correlation, suggesting that this signal is a quaternary carbon. By observing the HMBC experiment, this carbon presented correlations with the hydrogens H-3 (δ 3.03) and H-12 (δ 2.26) at 2J and 3J, respectively. Therefore, we opted to assign the signals at δ 71.9 and 70.6 to C-9 and δ C-2, respectively (Table 2). 

### 4.3. Biological Activity

As mentioned previously, the methanolic extracts obtained from M1 and M2 displayed activity against acute myeloid cell lines (Appendix A). In order to guide the chemical investigation, the fractions obtained from M2 were evaluated against a panel of acute myeloid and lymphoid leukemic cells (Figure 4A–D). Based on the cell viability of the dichloromethane and ethyl acetate fractions, the purification process proceeded with these fractions.

In order to examine the effect of Compounds (**7**), (**8**), (**9**), and (**10**) and the cell death mechanism, Kasumi-1 and Jurkat cells were selected. A reduction in cell viability was observed for all compounds in both lineages after 48 h (Figure 5A,B). Therefore, we constructed a concentration–response curve for Kasumi-1 and Jurkat cells after incubation (24 h) of compounds (**7**), (**8**), (**9**), and (**10**) (Figure 5C–F). Compound (10) presented a higher efficacy compared to the other compounds on target cells (the value of 50 μM EC_50_ was used for both) (Figure 5F). Staurosporine [20], a potent apoptosis inducer, was used as a positive control (Figure 5G). Additionally, compound (**10**) did not show a cytotoxicity effect on normal PBMC cells. 

Subsequently, EC_50_ values at 24 hrs were used to verify the cell death mode using Annexin V-FITC and 7-AAD staining. Kasumi-1 showed predominant apoptosis staining (Anx-FITC+/7 AAD-), whereas Jurkat showed predominant double staining (late apoptosis and necrosis-like) after 24 h (Anx-FITC+/7 AAD+) (Figure 6A–C). In order to corroborate the cell death effect of (**10**), flow cytometry was used to quantify the cleaved form of caspases 3 (activated form of caspases). The treatment with (**10**) increased cleaved-caspase-3 (Figure 7A–D) in both lineages. Furthermore, treatment with (**10**) produced a reduction in cell proliferation (Figure 8A–E) and cell cycle arrest in G0/G1 after 72 h of stimulus (10 μM). At this concentration, we did not observe a reduction in cell viability. An important activity in the antitumor response in acute myeloid leukemia is the ability of compounds to induce myeloid differentiation. Thus, the differentiation status was determined by the expression of mature myeloid marker CD11b after 72 h. Compound (**10**) did not produce an increase in the expression of CD11b (Figure 8F).

## 5. Discussion

Although recently [8,21], integrated approaches to molecular dereplication have become powerful and efficient strategies for discovering bioactive metabolites [22,23,24,25,26,27]. In addition, such tools promote the expansion of fungal and plant chemodiversity. Thus, recently, we have observed an expressive number of reports with the prospect of reinvestigating known matrices [28,29]. This dynamic offers new opportunities for accelerating access to new drug candidates, in particular, with cytotoxic potential [30,31,32,33]. In this sense, if we especially consider the *Penicillium* genus, our recent survey on the anti-leukemia potential of 76 compounds related to this group of fungi [2] showed that different classes of natural products present anti-leukemia effects, such as alkaloids (25% of the total), peptides/proteins (9%), polyketides (57%), and terpenoids (9%). 

Among the compounds presented in Table 1, some of these have already had their anti-leukemia activities reported. Emodin (**3**) showed a strong uncoupling effect on mitochondrial respiration by inhibiting L1210 cells with an IC_50_ of 35 μM [34]. Prior to our results, compound (**10**) showed potent activity against the HL-60 lineage, also an acute myeloid leukemia cell line (IC_50_ = 9.34 μM) [35]. 7-chloroemodin (**8**) showed inhibitory activity against ACL, an ATP citrate lyase (IC_50_ = 12.6 μM) [36]. This enzyme is essential for the generation of acetyl-CoA, which is a fundamental precursor for the biosynthesis of fatty acids and cholesterol. ACL was overexpressed in cancer cells and siRNA silencing impaired the proliferation of cancer cells. In this context, this substance presents good cytotoxic perspectives against tumorigenic cell lines. However, 2-Chloro-1-methyl-trihydroxy-6hydroxy-6methyl)anthraquinone (**7**) was inactive (Alamar Blue, 72 hrs) against Jurkat, K-562, and Hela [37]. Although some of these compounds have already been reported as potential anti-leukemia agents, we decided to purify them since previous studies carried out with *P. setosum* present incipient chemical data [38]. Thus, we are providing the first report of six compounds isolated and fully characterized from this fungal species.

## 6. Conclusions

Our dereplication approaches allowed the description of 15 annotated/dereplicated compounds originating from *P. setosum* CMLD 18 growth in different culture media (M1-M6). Six compounds were isolated, which enabled their structure confirmation. The three isolated chlorinated anthraquinones (**2**), (**7**), and (**8**), as well as the diketopiperazine (**9**), have not been described for the genus *Penicillium* before, representing a contribution to the chemical description of both genus and species. Moreover, the isolation of (**9**) and the NMR 1D and 2D experiments allowed the unequivocal assignment of the NMR shifts for this compound. Based on our findings, it is also possible to suggest that this fungus has a biosynthetic apparatus able to produce chlorinated metabolites, as it was indicated through the dereplication analysis. Importantly, this work also demonstrated Fellutanine C (**10**) cytotoxic effect, inducing cell death with apoptotic features and cell arrest in Jurkat and Kasumi-1. This is the first description regarding anti-leukemia activity for this diketopiperazine up to date. Moreover, on the basis of the obtained diketopiperazines, molecular families in *P.setosum* CMLD 18 extracts, unknown compounds of the same group are likely biosynthesized by the microorganism. Due to the cytotoxic activity observed for Fellutanine C, further bioassays with other diketopiperazines produced by this fungus may unveil other metabolites with anti-leukemia potential.

## Figures and Tables

**Figure 1 metabolites-13-00023-f001:**
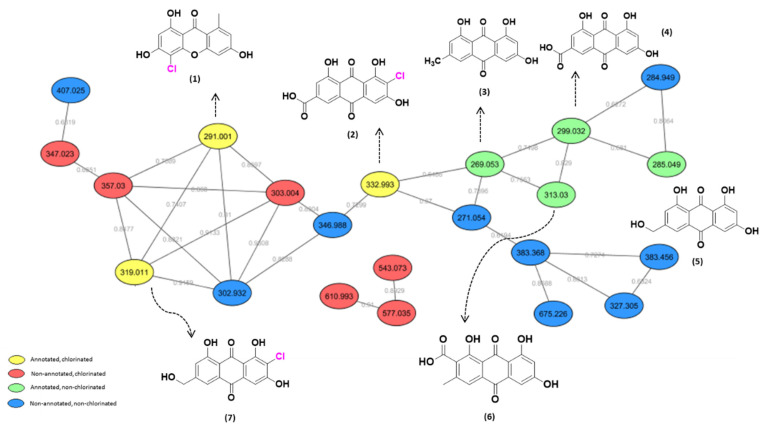
Xanthones and anthraquinones molecular family obtained via GNPS from fungal extracts prepared from *P. setosum*. Yellow nodes: chlorinated substances annotated by the platform; red nodes: not annotated by the platform but identified as chlorinated substances through the isotopic pattern; green nodes: non-chlorinated compounds annotated by the platform; blue nodes: compounds that were not annotated and did not have a typical fragmentation pattern of chlorinated substances.

**Figure 2 metabolites-13-00023-f002:**
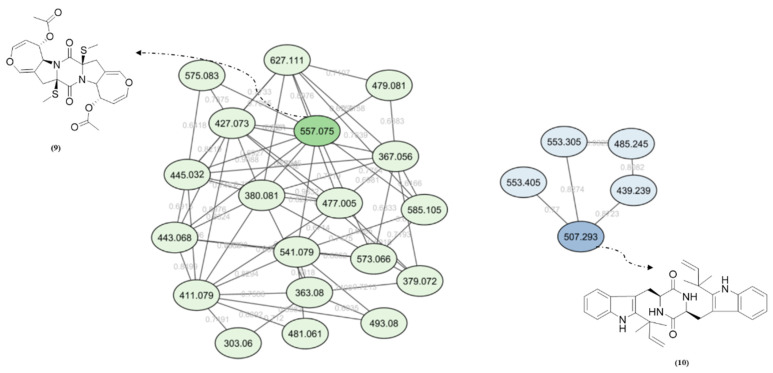
Diketopiperazine’s molecular family obtained via GNPS for fungal extracts prepared from *P. setosum*.

**Figure 3 metabolites-13-00023-f003:**
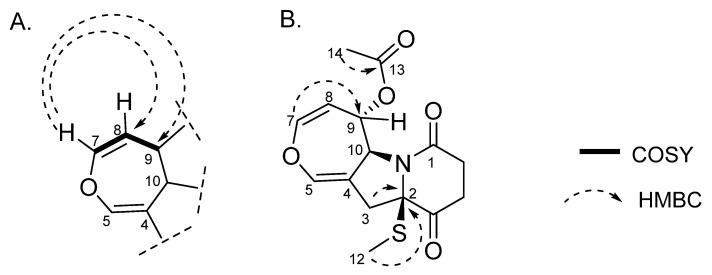
Main observed HMBC and COSY correlations for (9). (**A**) The correlation between the sign at δ 6.29 and H-8 (δ 4.69)/H-9 (δ 5.80) and δ 6.29 with C-8 and C-9 can be visualized through analysis of COSY and HMBC experiments, respectively. (**B**) The correlation of hydrogen at δ 2.07 with the carbonyl at C-13 and between the methyl hydrogens H-12 (δ 2.26) and C-2 (Figure 3B) are observed at HMBC experiments. All HMBC correlations are from protons to carbon (^1^H−^13^C).

**Figure 4 metabolites-13-00023-f004:**
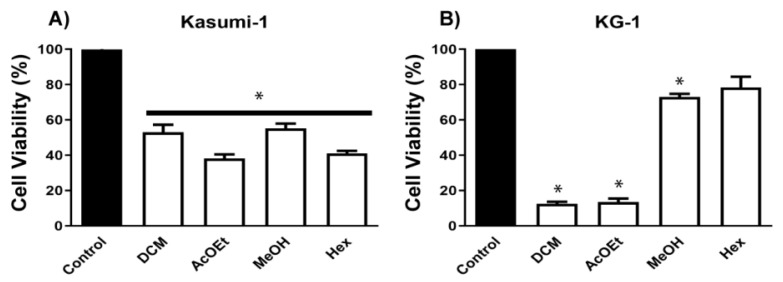
Cell proliferation and cytotoxic screening of fractions obtained from *P. Setosum*, cultivated in rice. (**A**) Kasumi-1, (**B**) KG-1, (**C**) Jurkat, and (**D**) MOLT-4 lineages were stimulated with different fractions at 100 μg/mL for 48 h. Cell viability was assessed using a resazurin assay. These results are the mean ± SEM of three independent experiments performed in triplicate. * *p* < 0.05. Statistical analysis was performed against a control (unstimulated sample). ANOVA test was followed by Dunnett’s post hoc test.

**Figure 5 metabolites-13-00023-f005:**
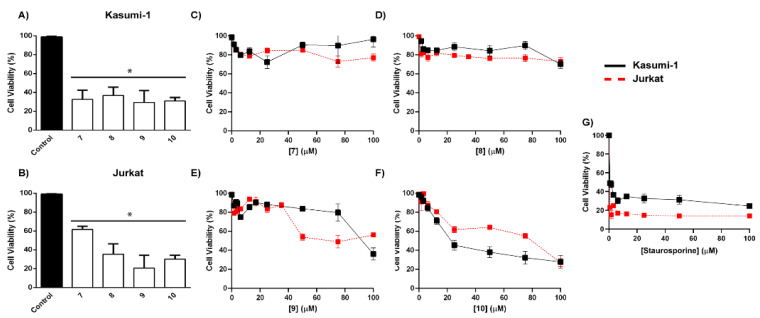
Cytotoxic screening and cell proliferation of isolated compounds obtained from *P. setosum* in leukemic cells. (**A**) Kasumi-1 and (**B**) Jurkat cells were stimulated with different compounds at 100 μM for 48 h. (**C**–**F**) A concentration–response curve on Kasumi-1 and Jurkat cell lines or normal PBMC were obtained after 24 h of treatment with isolated compounds or (**G**) Staurosporine (positive control). Cell viability was assessed using a resazurin assay. These results are the mean ± SEM of three independent experiments performed in triplicate. * *p* < 0.05. Statistical analysis was performed against a control (unstimulated sample). ANOVA test was followed by Dunnett’s post hoc test.

**Figure 6 metabolites-13-00023-f006:**
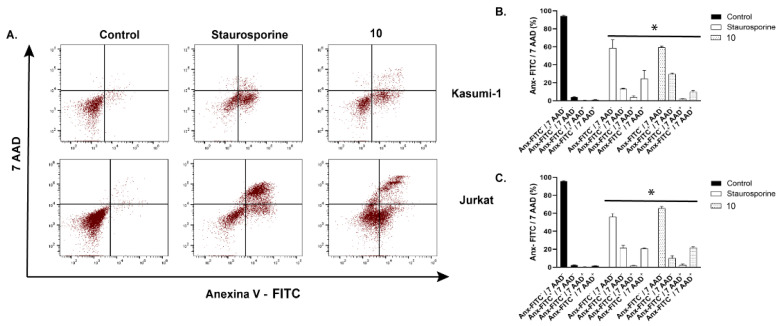
Quantitative analysis of cell death by compound (**10**) in leukemic cells. Kasumi-1 (50 μM) and Jurkat (50 μM) were stimulated with compound (**10**) for 24 h. The rate of apoptotic cells was assessed with annexin V-FITC/7-AAD staining. (**A**,**B**) Representative flow cytometry dot plots. (**C**) Quantitative measurements of cell death in Kasumi-1 and Jurkat. These results are the mean ± SEM of three independent experiments performed in duplicate. * *p* < 0.05. Statistical analysis was performed against a control (unstimulated sample). ANOVA test was followed by Dunnett’s post hoc test.

**Figure 7 metabolites-13-00023-f007:**
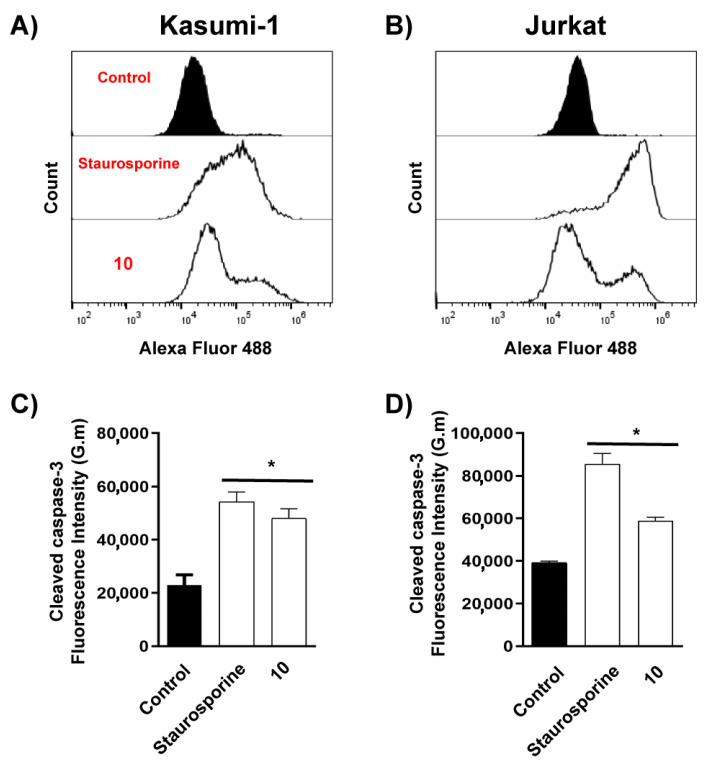
Activation of caspase-3 by compound (**10**) in leukemic cells. Kasumi-1 (50 μM) and Jurkat (50 μM) were stimulated with compound (**10**) or Staurosporine for 24 h. (**A**,**B**) Typical flow cytometry histogram; (**C**) Kasumi-1; (**D**) Jurkat cell lines. These results are the means ± SEM of three independent experiments performed in triplicate. * *p* < 0.05. Statistical analysis was performed against a control (unstimulated sample). ANOVA test was followed by Dunnett’s post hoc test.

**Figure 8 metabolites-13-00023-f008:**
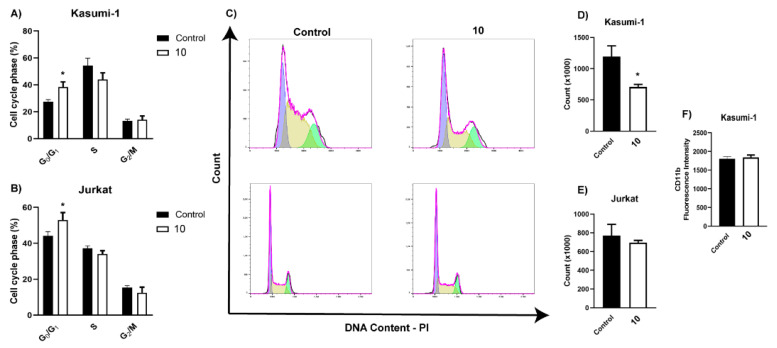
Compound (**10**) induces cell cycle arrest but does not promote myeloid differentiation in Kasumi-1 cells. Kasumi-1 and Jurkat were treated with compound **10** (10 μM) for 72 h. Cell cycle analysis was performed by flow cytometry using PI. (**A**,**B**) Percentages of cell cycle distributions; (**C**) typical flow cytometry histograms. Employed software excludes values of sub-G0/G1 population for quantifying DNA cell cycle; (**D**,**E**) cell count; (**F**) quantification of mature myeloid marker expression CD11b. These results are the means ± SEM of three independent experiments. * *p* < 0.05. ANOVA test was followed by Dunnett’s post hoc test or Students *t*-test.

**Table 1 metabolites-13-00023-t001:** *P. setosum*’s annotated and isolated compounds, their structures, detected masses, and the culture media of the fungal growth.

Compounds	Structure	*m/z*	Found in the Culture Media
4-chloro-norliquexanthone (**1**)	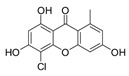	291.001	M6
2-chloroemodic acid (**2**) *	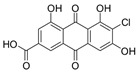	332.993	M1, M2, M4
emodin (**3**)	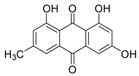	269.053	M1, M5
emodic acid (**4**)	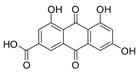	299.032	M1, M4, M5, M6,
citreorosein (**5**)	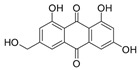	285.049	M2, M5, M6
endocrocin (**6**)	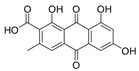	313.03	M6
2-chloro-1,3,8-trihydroxy-6-(hydroxymethyl)anthracene-9,10-dione (**7**) *	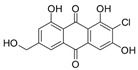	319.011	M1, M2, M4, M5, M6
7-chloroemodin (**8**) *	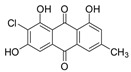	303.008	M2, M6
bisdethiobis(methylthio)acetylaranotine (**9**) *	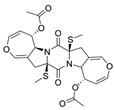	557.1028	M2
fellutanine C (**10**) *	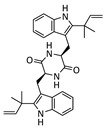	507.2750	M1, M2
canrenone (**11**)	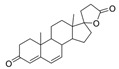	339.214	M6, M2, M5
tricin (**12**)	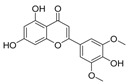	329.083	M2
5,6,2′-trimethoxyflavone (**13**)	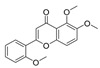	311.084	M2
hydroquinidine (**14**)	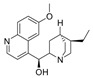	325.196	M2, M4, M5,
4-methyl-5,6-diihydro-2H-pyran-2-one (**15**) *	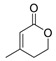	113.0601	M2

* Isolated compounds from *P. setosum* CMLD 18.

**Table 2 metabolites-13-00023-t002:** ^1^H and ^13^C NMR values obtained for (**9**).

Position	Compound (9)	
	δ_H_ (Multiplicity, J (Hz))	δ_C_
**1**	-	164.5
**2**	-	70.6
**3**	2.98–3.10 (*m*)	40.6
**4**	-	109.6
**5**	6.57 (*d*, 8.2)	137.8
**7**	6.29 (*dd*, 8.3 and 2.2)	139.8
**8**	4.69 (*dd*, 8.2 and 1.3)	105.9
**9**	5.80 (*bd*, 8.1)	71.9
**10**	5.17 (*bd*, 7.6)	60.4
**12**	2.26 (*s*)	14.8
**13**	-	170.2
**14**	2.07 (*s*)	21.1

## Data Availability

The data presented in this study are available in the main article and the Appendix A.

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
