# Peer review of "Chemodiversity and Anti-Leukemia Effect of Metabolites from *Penicillium setosum* CMLD 18"

_metabolites, 2022, doi:10.3390/metabo13010023_

Round 1

Author Response

Authors: dear colleague, thank you for your valuable contribution for improving our manuscript. In the new version, all changes are highlighted in yellow.

Major question:

  1. About the compounds 9 and 10, it would be interesting for the authors to present the absolute configuration of them. The compound 9 was obtained in a few amounts and it would be necessary to compared with literature data. However, compound 10 was obtained in a great amount and in this case the proposition of an absolute configuration is mandatory.

Authors:

Regarding the absolute configurations of compounds 9 and 10, we believe that it is not necessary to carry out experiments to propose the stereochemistry of these compounds, since they are not new in the literature. The characterization of the structures was performed in comparison with previously reported data. However, for compound 10, a-D experiment was performed to determine its specific rotation, as can be seen in section 3.4 (compounds isolation), which is consistent with data reported by Kozlovsky, A.G et al. (reference 18)

Although we obtained a reasonable amount of this substance, it is necessary to point out that the sample was submitted to many cytotoxic assays to determine its biological effect. In the end, there is no amount enough to carry out additional experiments. We are very sorry for it.

Minor questions:

  1. The number of authors for this work is too high. Please explain the main role of each authors;

Authors: the role of each author is described in the section Author Contributions. Unfortunately, or fortunately, depending on the point of view, in Brazil, science is not performed individually. As mentioned by the reviewer 2 “The work appears extensive and conducted with appropriate methods whose results support the rationale and are in agreement with the title of the paper”. Therefore, based on the interdisciplinary approaches employed by us, naturally we needed different people involved in the team.

  1. B) What was the final concentration of the compounds obtained in DMSO? This information is important and should be mentioned in the text.

Authors: As you can find in the section 3.1.1 (stock solutions), in all experiments the maximum concentration of DMSO did not exceed 0.25%. This concentration does not affect these lineages.

  1. C) Was the collection of the plant authorized by competent agencies?;

Authors: S. glutinosa is widely used as an ornamental plant, especially in living fences or in plantations as a windbreak barrier (Weniger et al, 2001). Therefore, its seedlings were purchased in the online market for cultivating in the university garden (23°43'10.7"S 46°37'38.5"W). After six months of planting, we performed the isolation of its endophytic fungi, including P. setosum.

Weniger, B. et al (2001). Antiprotozoal activities of Colombian plants. J. Ethnopharmacology, 78, 193-200, doi: 10.1016/S0378-8741(01)00346-4

  1. D) How many Erlenmeyer flasks were used in the experiments of Penicillium setosum cultivation in hominy (M1) in CYB (M3) and YES (M4)?;

Authors: It was included in the new version.

  1. E) Employing hexane, dichloromethane, ethyl acetate, and methanol as eluents in gradient mode… Specify the gradient used;

Authors: It was included in the new version (section 3.4).

  1. F) Extracts from M1 and M2 presented activity against acute myeloid cell line… Couldn't compounds from culture media infer biological activity?

Authors: all conditions for growth and maintenance of cell lines followed the standards recommended of ATCC cell bank (https://www.atcc.org/products/ccl-246#detailed-product-information). To exclude the possibility of interferences, a positive and negative controls were employed in all experiments. For additional information:

Torquato, H.F.V., et al., A canthin-6-one derivative induces cell death by apoptosis/necroptosis-like with DNA damage in acute myeloid cells. Biomed Pharmacother, 2022. 145: p. 112439.

  1. G) This approach allowed us to annotate 13 compounds produced… But notice this previously phrase: The combination of these approaches led to the annotation of fifteen

Authors: It was changed in the new version.

Reviewer 2 Report

* The abstract: reformulate it. I would like to see an abstract organized according to: Background, objective, methods, results and discussion
* Introduction: The introduction doesn't  identify the problem that is being addressed in the manuscript and develops and states the purpose of the manuscript. Please overview your (proposed) approach more accurately. Please also provide a rationale behind using the proposed approach, i.e., what are its benefits in comparison to alternative approaches. What are the main limitations of alternative approaches in comparison to the proposed approach? What is the gap that you aim to address with your approach?
* Quality of figures is so important too. Please provide some high-resolution figures. Some figures have a poor resolution.
* There are still some mistakes in grammar and misprints, the authors should carefully check this manuscript.
* The authors don't discuss the limitations of this study correctly.  Please add a section (discussion).

* References: please add more references to give credit to the introduction. 

Author Response

Authors: dear colleague, thank you for your valuable contribution for improving our manuscript. In the new version, all changes are highlighted in yellow.

  1. The abstract: reformulate it. I would like to see an abstract organized according to: Background, objective, methods, results and discussion.

Authors: The abstract was fully reformulated.

  1. B) Introduction: The introduction doesn't  identify the problem that is being addressed in the manuscript and develops and states the purpose of the manuscript. Please overview your (proposed) approach more accurately. Please also provide a rationale behind using the proposed approach, i.e., what are its benefits in comparison to alternative approaches. What are the main limitations of alternative approaches in comparison to the proposed approach? What is the gap that you aim to address with your approach?

Authors: In this version, we included a paragraph for clarifying these points.

  1. C) Quality of figures is so important too. Please provide some high-resolution figures. Some figures have a poor resolution.

Authors: The figures resolution was increased (300 dpi).

  1. D) There are still some mistakes in grammar and misprints, the authors should carefully check this manuscript.

Authors: the first version was submitted to Cambridge Proofreading for language review in accordance with the attached certificate. Anyway, the text was revised again.

  1. E) The authors don't discuss the limitations of this study correctly.  Please add a section (discussion).

Authors: In this version, we included a discussion. We really appreciate your observation.

  1. F) References: please add more references to give credit to the introduction. 

Authors: In this version, we included more references.

Reviewer 3 Report

In this work, Authors have performed a chemical and biological study of Penicillium setosum CMLD18, an endophyte from Swinglea glutinosa. They describe the extraction, purification, NMR analysis of six substances, then tested against leukemia cell lines. Based on the results, Authors state that their work increases the knowledge about the P. setosum chemical profile and its biological potential as anti-leukemia properties.

The work appears extensive and conducted with appropriate methods whose results support the rationale and are in agreement with the title of the paper.

However, some points need to be improved and/or clarified.

1.Authors state, already in the abstract, that since of the assayed metabolites, diketopiperazine fellutanin C induced cellular death against Kasumi-1, a human leukemia cell line, as well as good selectivity for it, displaying promising activity.

This point is not clear since also the other three cell lines belong to the same tumour type. In addition, usually, selectivity should be used when a compound believed to have anticancer action is active or more active against cancel cells but not or less against normal or non-tumorigenic cells.

Perhaps these compounds could also be tested on normal cells.

2.Some different responses among the four cell lines could be ascribed to the different cell culture media. In this regard, KG-1 cells were cultured in Iscove's modified Dulbecco's medium (IMDM) supplemented with 20% FBS, whereas the other lineages were maintained in RPMI 1640 supplemented with 10% FBS. Therefore, in addition to the double amount of FBS, the two media also differ greatly in the amount of many of their components.

Is this perhaps the reason why the apoptosis data in KG-1 cells are not shown?

3. I wonder why, despite the data in Figs 7 and 8, the diagrams in panel 8C do not show cells in the sub-G1 hypodiploid phase of the cell cycle after PI staining?

Author Response

Authors: dear colleague, thank you for your valuable contribution for improving our manuscript. In the new version, all changes are highlighted in yellow.

  1. Authors state, already in the abstract, that since of the assayed metabolites, diketopiperazine fellutanin C induced cellular death against Kasumi-1, a human leukemia cell line, as well as good selectivity for it, displaying promising activity.

This point is not clear since also the other three cell lines belong to the same tumour type. In addition, usually, selectivity should be used when a compound believed to have anticancer action is active or more active against cancel cells but not or less against normal or non-tumorigenic cells. Perhaps these compounds could also be tested on normal cells.

Authors: leukemia cells employed in this study are from different moment of disease with specific particularities and mutations. The compound (10) was also assayed against peripheral mononuclear cells obtained from healthy donor. This information is described in the section 3.1.3 and also highlighted in the section 4 (Biological activity).

2.Some different responses among the four cell lines could be ascribed to the different cell culture media. In this regard, KG-1 cells were cultured in Iscove's modified Dulbecco's medium (IMDM) supplemented with 20% FBS, whereas the other lineages were maintained in RPMI 1640 supplemented with 10% FBS. Therefore, in addition to the double amount of FBS, the two media also differ greatly in the amount of many of their components.

Is this perhaps the reason why the apoptosis data in KG-1 cells are not shown?

Authors: the growth conditions followed the criteria adopted by the cell bank (https://www.atcc.org/products/ccl-246#detailed-product-information). The KG-1 strain, in particular, has indicated the IMDM culture medium by has a fast-growing strain. We decided to select only two cell lines for further investigation because Kasumi-1 cells (AML) present frequent cytogenetic abnormalities in the population and Jurkat cell is an acute T-cell leukemia patient. Therefore, we selected two strains obtained from hematological diseases with distinct genetic and clinical characteristics.

  1. I wonder why, despite the data in Figs 7 and 8, the diagrams in panel 8C do not show cells in the sub-G1 hypodiploid phase of the cell cycle after PI staining?

Authors: Figure 7 shows the cleavage of caspase-3 in all cell population, DNA label was not performed in this experiment. Figure 8 DNA label showed the cell cycle analysis using PI. The software excludes values of sub-G0/G1 population for clear quantifies DNA cell cycle. Figure 6 is represented by labeling with Annexin V-FITC/ 7-AAD, where is possible differentiate apoptotic cells, which to some extent activate DNA damage mechanisms. For further information:

Larsen, B.D. and C.S. Sorensen, The caspase-activated DNase: apoptosis and beyond. FEBS J, 2017. 284(8): p. 1160-1170.

Vieira Torquato, H.F., et al., Canthin-6-one induces cell death, cell cycle arrest and differentiation in human myeloid leukemia cells. Biochim Biophys Acta Gen Subj, 2017. 1861(4): p. 958-967.

Reviewer 4 Report

Comments:

1. The study depicts a fungus identification done − Penicillium setosum CMLD18, the phylogenetic analysis of the identified Penicillium setosum was conducted. However, the authors are missing some information about this fungus. Please provide in the revised manuscript.

- The pictures of the fungus Penicillium setosum both macroscopic and microscopic.

- ITS region of the rDNA of the fungus Penicillium setosum.

2. The NMR spectrum of isolated compounds in the Supplemental material should be rearrange in sort of compound’s number to make easier for the reader to follow your study.

3. Figure 3C are revealed all key HMBC correlations of compound 9. Thus, fig 3B should deleted. Also, the legend of subfigures in Fig 3 is required. Please add.

4. Section 3.2. Where is the plant material Swinglea glutinosa collected? Please add. Also, state the sampling coordinate (if possible).

5. There are some minor errors need to revise.

- Abstract section: “RUTACEAE” --- “Rutaceae”

- Section 3.4: C18 Luna – Phenomenex - “250 x 10.0 MM” --- “250 x 10.0 mm”

Author Response

Authors: dear colleague, thank you for your valuable contribution for improving our manuscript. In the new version, all changes are highlighted in yellow.

1) The pictures of the fungus Penicillium setosum both macroscopic and microscopic.

Authors: the complete morphological characterization of fungal strains is a common practice in taxonomic papers in the occasion of description of new taxa. Since we worked with a known Penicillium specie that was recently fully described in an open access publication (George et al., 2019), we do not find it relevant to provide pictorial data on the morphological features of the isolate (microscopic pictures). The macroscopic pictures of Penicillium setosum CMLD 18 were added in Supplementary Material (figure S2).

2) ITS region of the rDNA of the fungus Penicillium setosum.

Authors: the ITS region of rDNA is considered the official barcode for fungal identification. However, it is well-known that it may provide poor resolution for species delimitation among close related taxa in many fungal genera, including Penicillium, and this limitation can be resolved by adopting a secondary DNA barcode, such as β -tubulin (Visage et al. 2014). In the present work, we have adopted the partial sequence of the β -tubulin as a gene marker for phylogenetic analysis and identification of strain CMLD 18 as Penicillium setosum. The provided phylogenetic tree clearly showed that β -tubulin is sufficient for delimitation of P. setosum from phylogenetically related taxa, as was the case when P. setosum was firstly described (George et al. 2019a).

Our approach for the identification of P. setosum CMLD 18 was based on the recommendations provided by Visage et al. (2014) in an extensive monograph of the genus Penicillium. In the words of the aforementioned authors: "Unfortunately, for Penicillium and many other genera of ascomycetes, the ITS is not variable enough for distinguishing all closely related species ... Because of the limitations associated with ITS as a species marker in Penicillium, a secondary barcode or identification marker is often needed for identifying isolates to species level...  we propose the use of β -tubulin (BenA) as the best option for a secondary identification marker for Penicillium."

George, T. K., Houbraken, J., Mathew, L., Jisha, M. S. (2019a). Penicillium setosum, a new specie from Withania somnifera (L.) Dunal. Mycology, 10(1), 49-60.

Visagie, C. M., Houbraken, J., Frisvad, J. C., Hong, S. B., Klaassen, C. H. W., Perrone, G., Seifert, K. A., Varga, J., Yaguchi, T., Samson, R. A. (2005). Identification and nomenclature of the genus Penicillium. Studies in Mycology, 53(1), 53-62.

3) The NMR spectrum of isolated compounds in the Supplemental material should be rearrange in sort of compound’s number to make easier for the reader to follow your study.

Authors: the NMR spectra in the Supplementary material were arranged following the compound´s number.

4) Figure 3C are revealed all key HMBC correlations of compound 9. Thus, fig 3B should deleted. Also, the legend of subfigures in Fig 3 is required. Please add.

 Authors: it was reformulated in the new version.

5) Section 3.2. Where is the plant material Swinglea glutinosa collected? Please add. Also, state the sampling coordinate (if possible).

Authors: S. glutinosa is widely used as an ornamental plant, especially in living fences or in plantations as a windbreak barrier (Weniger et al, 2001). Therefore, its seedlings were purchased in the online market for cultivating in the university garden (23°43'10.7"S 46°37'38.5"W). After six months of planting, we performed the isolation of its endophytic fungi, including P. setosum.

Weniger, B. et al (2001). Antiprotozoal activities of Colombian plants. J. Ethnopharmacology, 78, 193-200, doi: 10.1016/S0378-8741(01)00346-4

6) There are some minor errors need to revise.

- Abstract section: “RUTACEAE” --- “Rutaceae”

- Section 3.4: C18 Luna – Phenomenex - “250 x 10.0 MM” --- “250 x 10.0 mm”

Authors: it was changed in the new version.

Round 2

Reviewer 1 Report

Accept in present form

Author Response

Dear reviewer,

Thank you for your valuable collaboration. 

Best regards

Lívia

Reviewer 2 Report

I would like to thank the authors for their great job. 

Author Response

(The authors gave the same response as above.)

Reviewer 3 Report

The authors replied to the reviewer on the questions posed, but the explanations relating to points 2 and 3 do not appear to have been inserted in the text. If so, I would like to add a few sentences to clarify these points.

Author Response

Dear reviewer,

Thank you for your collaboration. In this round, we highlighted your requests in light blue. Please, take a look on the section 3.1.2 Cell cultures and also on the legend of figure 8. 

Best regards,

Lívia